# Daily Life Patterns, Psychophysical Conditions, and Immunity of Adolescents in the COVID-19 Era: A Mixed Research with Qualitative Interviews by a Quasi-Experimental Retrospective Study

**DOI:** 10.3390/healthcare10061152

**Published:** 2022-06-20

**Authors:** Ji-Eun Yu, Denny Eun, Yong-Seok Jee

**Affiliations:** 1Department of Physical Education, Korea University, #145 Anam-ro, Seongbuk-gu, Seoul 02841, Korea; jieun12155@naver.com; 2Department of Physical Education, Yonsei University, #50 Yonsei-ro, Seodaemun-gu, Seoul 03722, Korea; 3Department of Leisure and Marine Sports, Hanseo University, #1 Hanseo-ro, Haemi-myeon, Seosan 31962, Korea

**Keywords:** adolescents, daily life pattern, depression, physical fitness, NK cell, T cell

## Abstract

Background: This study investigated the daily lifestyle changes, prevalence of psychological depression, physical health status, and immunity of adolescents in Korea resulting from increased isolation and social restriction due to the COVID-19 pandemic. Materials and Methods: All subjects included 17-year-old male adolescents. A total of 117 subjects were assigned to one of four groups according to the degree of depression based on item #6 in the Center for Epidemiologic Studies Depression (CES-D) questionnaire as follows: no-depression group (NDG, *n* = 71; 61.0%), low-depression group (LDG, *n* = 23; 19.0%), moderate-depression group (MDG, *n* = 15; 13.0%), and high-depression group (HDG, *n* = 8; 7.0%). This study analyzed the data using quantitative and qualitative methods to understand how the COVID-19 pandemic affects adolescents’ daily lives, psychophysiological conditions, and immune function. Results: This study found that the COVID-19 pandemic significantly affects the daily lifestyle pattern, psychophysical condition, and immunocytes of adolescents. In terms of depression, 39.0% of adolescents felt depressed, and 7% of them felt depressed almost every day. Overall, HDG considered themselves unhealthy and felt prone to immune diseases, such as colds. HDG were prone to sleep late, eat more frequently, and work out less. Regarding physical fitness factors, the cardiorespiratory endurance, strength, and power of HDG were significantly lower than those of NDG, LDG, and MDG. Moreover, HDG had the worst body composition, including the lowest muscle mass. Finally, natural killer (NK) cells and T cells were significantly different among groups, with the levels in HDG being significantly lower than those of the other three groups. Conclusions: Since the COVID-19 pandemic negatively affects the daily lives, psychophysical conditions, and immunocytes of adolescents, there is an urgent need to create and provide solutions to adolescents with depression though the number of subjects is few.

## 1. Introduction

People from around the world are struggling to deal with the many changes caused by the coronavirus disease 2019 (COVID-19) pandemic that broke out in 2019 [1]. This refers to an infectious disease caused by severe acute respiratory syndrome coronavirus 2 [2]. The number of coronavirus infected patients in Korea increased steadily to 342,446 during the course of this study. On 17 March 2022, 621,328 people were reported to have been infected. This pandemic has caused not only physical problems but also psychological problems, such as depression, anxiety [3,4,5], and stress [6]. Fear of the spread of coronavirus is having a significant impact on society as a whole. The psychological effects of anxiety and fear about the possibility of infection or death have led to social distancing, self-isolation [7], closure of public institutions, and business hour restrictions in Korea. According to some studies, it is reported to have caused a number of negative consequences on daily social activities [8]. In particular, depression, anxiety, and negative emotions can lead to maladaptation that adversely affects daily life [9,10].

Psychological depression and anxiety caused by the pandemic have negatively impacted people’s physical health and daily lives. In the case of adolescents, it is reported that this phenomenon worsens when social networks are not established and when physical activity, as well as daily life, cannot be freely performed [11]. Moreover, the pandemic makes it difficult for adolescents to meet friends, teachers, and relatives, which can impede personality formation and social development. In addition, self-isolation and low physical activity is causing a deterioration of physical fitness, immunity [12], and overall health [13,14]. Psychological depression, anxiety, and decreased physical fitness are more severe in adolescents than in other age groups and are rapidly spreading in the younger generation [11].

Since depression has a vicious cycle of avoidance—isolation—self-criticism—perfectionism—despair, breaking this cycle is crucial. Depression can lead to suicidal tendencies, which are more likely to occur in adolescents who are accustomed to living an active lifestyle [15]. Specifically, self-isolation interferes with daily life and reduces the level of physical activity, which may increase the possibility of depression and chronic diseases related to immunotoxicity [12,16,17,18,19,20].

It is important to identify the degree of depression that can interfere with positive learning attitudes, healthy physical activity, and social adaptation. However, the focus of governments and school authorities has been entirely on preventing further infections. In addition, it is difficult to investigate the daily life of students and their mental health. In this context, this study sought to improve the psychophysiological health of adolescents and, more specifically, to find out the effects of the pandemic on the depression levels in adolescents and whether there are differences in immunocytes function and daily life according to the degree of depression.

## 2. Materials and Methods

### 2.1. Experimental Design

This was a quasi-experimental retrospective study that investigated both quantitative research analysis and qualitative interview content to understand how the COVID-19 pandemic affects adolescents’ daily lives, psychophysiological conditions, and immunocyte function. All data in this study were acquired from 6 September 2021 to 6 October 2021. The participants we recruited for this study were male middle-school students. Since the participants’ conditions before the COVID-19 pandemic could not be quantitatively grasped, they were asked during the qualitative interview to make a comparison of their daily life and psychophysiological conditions before and during the COVID-19 pandemic.

### 2.2. Ethical Approval

The Korean government looked at the trend of newly infected COVID-19 cases and adjusted restrictions accordingly. In line with this, school authorities decided whether in-person classes would take place and required students to wear masks at school. For this study, a research plan was made according to what we believed was likely to be the circumstances at the start of the study. Upon approval from the IRB committee, the experiment was carried out. Prior to the study, the participants and their parents received detailed explanations regarding the study procedures and were then asked to complete a questionnaire, have a blood sample taken, and complete a physical fitness test. Although the tests performed in this study were compulsory tests performed by schools, we obtained informed consents from their parents, as students were considered a vulnerable group. All the subjects, including the researchers, did not know which group the participants belonged to. This study started after receiving IRB approval (2-1040781-A-N-012020 085HR), and the data of all participants were kept confidential and used for research purposes only. After thoroughly checking the social distancing status when students were able to go to school, only students who had normal body temperature and no COVID-19 symptoms gathered in the experimental research center. Daily life patterns that can affect the mind and body included self-health and disease status, sleep and wake-up time, meal frequency, and exercise. For the questionnaire investigating psychological state, CES-D was used to gauge the depression levels of the participants, and physical condition was measured as a health-related physical fitness component. The natural killer (NK) and T cells were identified through blood tests.

### 2.3. Subjects

This study recruited 128 male students in their third year of an all-boys middle school in Korea who expressed their intention to participate in this study. All subjects were 17 years old. The sample size using G*Power (v. 3.1.9.7, Heinrich-Heine-University Software, Germany) was calculated by adding the number of subjects required in the analysis of covariance (ANCOVA) considering a priori the effect size of f^2^ (V) = 0.40 (large effect), *α* error probability = 0.05, power (1-*β* error probability) = 0.95, number of groups = 4, and numerator difference = 1. The number of samples obtained by considering the experimental design of this study as G*power was 84, but the number of samples was rather high, as there were 117 students who had to undergo mandatory health and physical examinations.

Students who were required to undergo a mandatory health and physical examination every year were included in this study. The inclusion criteria also required that the participants had not received treatment or medication known to affect mental status and body composition. Exclusion criteria consisted of having a history of impairment of a major organ system or a psychological disorder. Students who had or had been infected with the coronavirus or corona variant virus were also excluded from the study. It is thought that infection with the coronavirus causes significant changes in immune cell function, which can act as an extra variable not only in physical fitness but also in psychological aspects, so it was excluded from this study. 

The Center for Epidemiologic Studies Depression (CES-D) questionnaire [21] was used to identify the students’ level of depression. The content of item #6 asked about “the number of days when you felt depressed in the past week”. If a subject marked, “I rarely felt depressed” or “I felt depressed for less than 1 day a week”, the subject was classified in the no-depression group (NDG). If a subject marked, “I felt depressed about 1~2 days a week”, the subject was classified in the low-depression group (LDG). If a student indicated that he felt depressed about 3~4 days a week, he was classified in the moderate-depression group (MDG). Lastly, if a student indicated that he felt depressed about 5~7 days a week, he was classified in the high-depression group (HDG).

As shown in Figure 1, 122 subjects were assigned to one of four groups according to the level of depression. Of the 72 subjects who were allocated to the NDG, one did not receive a physical fitness test. Of the 24 subjects who were allocated to the LDG, one did not sign a consent form. Of the 17 subjects allocated to the MDG, one did not complete a physical fitness test, and one chose not to continue. Of the 9 subjects who were allocated to the HDG, one did not complete a physical fitness test. Finally, a total of 117 participants were included in the study.

### 2.4. Measurement Methods

#### 2.4.1. Daily Life Pattern Questionnaire

A questionnaire with a simple scale composed of 7 items was developed to measure the daily life of the subjects. The questionnaire contents were structured so that questions could be easily answered by middle-school students and were revised and supplemented by two experts to record the subjects’ general lifestyles for one month. This Likert scale survey was produced and distributed through a social networking service (SNS). The participants completed the questionnaire and submitted the answers anonymously in the same location. The same questionnaire was distributed twice, collected, and analyzed for reliability. As shown in Appendix A of Appendix A, Question Q1 and Q2 of the 7 questions are related to health. Q1 asked how healthy the subjects felt, and Q2 asked about their immune health. The Cronbach’s α for Q1 and Q2 were 0.734 and 0.761, respectively. The three sleep-related questions (Q3~Q5) consisted of checking the times of going to sleep and waking up. Meal-related questions (Q6) were initially composed of the type of food the subjects ate in a day, the amount of food, when they ate, and how many times they ate. However, in consideration of the many details that the subjects need to remember, daily patterns related to meals were limited to only the number of meals per day. The exercise-related questionnaire was simplified to ask only about exercise frequency. The Cronbach’s α of the daily life patterns in this study was 0.711.

#### 2.4.2. CES-D Questionnaire Measures

This study was used by applying the CES-D questionnaire developed by Radloff [21]. The questionnaire scale consists of 20 items to measure symptoms related to depression. CES-D was also distributed through SNS and automatically collected in one place anonymously. The subjects were asked how they felt about the contents of the questions throughout the past 7 days. Questions 4, 8, 12, and 16 out of a total of 20 items were recorded as 3 points if a subject indicated “less than 1 day” or “none”. If a subject indicated “about 1 to 2 days”, 2 points were recorded; if a subject indicated “3 to 4 days”, 1 point was recorded, and if a subject indicated “about 5 to 7 days”, 0 points were recorded. The other items provided scores opposite to the above items. The score was the sum of the 20 questions with a possible range between 0 and 60 points. A score of 16 points or more was considered depressed. If answers to 4 or more questions were omitted, the CES-D question was not scored, and the entire questionnaire was excluded. The reliability of the questionnaire using Cronbach’s α was found to be 0.795.

#### 2.4.3. Qualitative Interview Measures

In this study, the interview and analysis process were done according to a study by Spradley [22] and Silverman [23]. When we first planned this study, the research model focused entirely on “quantitative research”, but we wanted to convey the direct feelings of the students going through the COVID-19 pandemic after completing the preliminary survey. The subjects who were interviewed included 8 adolescents in HDG who experienced severe psychological changes due to the COVID-19 pandemic. During the analysis process, the interviewers found the students who responded to the SNS questionnaire and asked the homeroom teacher to obtain their phone numbers. One of the researchers delivered a message to the student via SNS and asked how the interview was. If the interview contents sent by the students were similar, the contents similar to the situation were selected as the results of the study, and if they did not match the intention of the question, they were deleted. The collected and analyzed interviews were inserted in appropriate sections of the discussion. This study aimed to investigate the process of physical and mental changes that students experienced during the COVID-19 pandemic compared to before the COVID-19 pandemic. In addition to quantitative data analysis, additional interviews were conducted to supplement the quantitative research results. The questions for the interview were as follows.


*First, are there any changes in your daily life due to the COVID-19 pandemic?*



*Second, are there any mental changes caused by the COVID-19 pandemic?*



*Third, are there any physical changes caused by the COVID-19 pandemic?*



*Lastly, how have you dealt with those changes?*


#### 2.4.4. Immunocytes Measures

Blood samples were collected using BD vacutainer tubes (Becton Dickinson, Franklin Lakes, NJ, USA) to evaluate immunocytes after fasting for 10 h. After resting for 15 min, 5 mL of blood was collected from the antecubital vein of the subjects with a disposable syringe by a medical laboratory technologist. Blood samples were left at room temperature for 1 h and centrifuged at 1000 rpm for 15 min for the serum. After that, the immunocytes measures were conducted to evaluate the cellular immune function of the subjects, and the distribution and levels of leucocyte, T cells, and NK cells present in the subjects’ peripheral blood were confirmed. The distribution of CD3-positive T cells and CD56-positive NK cells was measured using flow cytometry [12]. Specifically, the percentage and absolute cell counts of peripheral blood cell subsets were analyzed as described below: 50 μL of blood was stained with anti-human antibodies against anti-human CD3-Fluorescein isothiocyanate (FITC; Cat No. 555339, BD) and CD56-Phycoerythrin (PE; Cat No. 555516, BD) from BD Biosciences (Franklin Lakes, NJ, USA). After incubation for 15 min at room temperature in the dark, the erythrocytes were lysed by adding 450 μL of FACS lysing solution to each test tube for another 15 min at room temperature in the dark. The variables were then analyzed using FACS Canto II (BD Bioscience) and Flowjo software (Treestar, Ashland, OR, USA) and are presented as percentages, as shown in Figure 2. Absolute cell counts of lymphocyte subsets were obtained using an automatic hematology analyzer (Sysmex Corp., Kobe, Japan).

#### 2.4.5. Body Composition Measures

In this study, height was measured in centimeters using a stadiometer. The body weight was measured in kilograms using a digital scale, which was measured in a standing position without shoes or other accessories while the subject was wearing clothes. Body mass index (BMI) refers to the value obtained by dividing the weight by the square of the height and is usually expressed in kg/m². In addition to BMI, muscle mass, fat mass, percent fat, and waist/hip ratio (WHR) were measured and analyzed using Inbody 320 (Biospace Co., Ltd., Seoul, Korea). This analyzer is a segmental impedance device that assesses the voltage drop in the upper and lower body. All subjects were asked to hold onto the handles and stand still for around 3 min. They were also asked to void 30 min prior to the assessment.

#### 2.4.6. Physical Fitness Measures

In this study, factors that require a certain amount of rest in the middle of measurement were taken into consideration, and flexibility, muscle strength, power, and cardiorespiratory endurance (cardioendurance) were measured in order. Flexibility was measured by a sit-and-reach test, muscle strength by grip strength test, power by standing long jump, and cardioendurance by 20 m shuttle run. For injury prevention, a warm-up exercise was performed before measurement, and the inspection, execution method, and precautions were explained. The flexibility test measured the degree of bending the upper body while in a sitting position with both legs straight. The subjects’ feet were checked to ensure they made complete contact with the measuring device (TKK1859, Takei Inc., Tokyo, Japan), while the researcher lightly pressed the knees to keep them straight when the subjects bent forward with their arms and hands stretched forward. The point where the subjects’ fingertips reached on the measuring scale in units of 0.1 cm was recorded if it was maintained for 2 s. The subjects were asked to perform this test twice, and the best score was recorded. Grip strength is a very important physical element because it is developed during adolescence. In this test, the digital dynamometer (TKK-5401, Takei Inc., Tokyo, Japan) was held so that the second joint of the finger was at a right angle. The subjects were asked to stand comfortably in an upright position and place their feet flat on the floor, shoulder-width apart. The subjects adjusted the width of the digital dynamometer to fit their hands, and measurements were performed twice on both sides. The highest score was recorded. The standing long jump was used to assess the whole body reflexes, which measures the ability to produce the maximum force in a short moment. The standing long jump power test can be measured with simple tools. The measurement was taken from the takeoff line to the point of landing with the heels. The best of two attempts was recorded to the nearest 0.1 cm. For the shuttle run, 20 m was prepared as the running distance. The recorders were instructed on how to record the pacer and to accurately give the starting signal. When the beep sounded, it was closely observed whether both feet completely passed the line and changed direction, whether they ran in a straight line without departing from the measurement line, and whether the entire person was moving in the same direction [24]. This test was completed once, and the maximum number of repetitions was recorded.

### 2.5. Sample Size and Data Analyses

The Shapiro–Wilk test was used to analyze the normality of distribution. Initially, the basic design of this study was designed to verify the differences by variable for the four groups by ANOVA. Prior to analysis, the Kruskal–Wallis test was used to verify and find physical conditions (age, height, and body weight) that could affect the purpose of the study. In the process, body weight was a significant difference between the groups. Therefore, a body weight was analyzed as a covariate for all variables according to ANCOVA model [25,26]. Effect size (partial *η²*) was calculated with Cohen’s d, which is equal to the mean difference of the groups divided by the pooled SD [27]. This study analyzed data as mean ± standard deviation using Microsoft Excel (Microsoft, Redmond, WA, USA), and SPSS (version 22.0; IBM Corp., Armonk, NY, USA) was used to analyze all data. Descriptive statistics and frequency analysis were performed to analyze the demographic results, including the subjects’ daily life patterns, and Cronbach’s *α* was calculated to check the reliability of the daily life pattern scale and CES-D questionnaire. In the general social science field, the acceptance criterion for accreditation of reliability is considered to be 0.6 or higher [28]. After the ANCOVA test was performed, a post hoc test among the four groups was done, and the Bonferroni adjusted *p*-value (<0.05/6 = 0.0167) was used for comparisons. Finally, the GraphPad Prism 9.3.1 (La Jolla, CA, USA) was conducted to compare the differences between the post hoc values. *p* ≤ 0.05 was considered to indicate statistically significant differences.

## 3. Results

### 3.1. Frequency Analysis for Depression

As shown in Figure 3, most subjects (61.0%) felt little or no depression, while the remaining 39.0% felt depression due to the pandemic. Among the subjects who thought they were depressed, 7.0% experienced a high level of depression.

### 3.2. Analysis of Daily Life Pattern

As shown in Table 1, although there was no significant difference among groups in Q3 and Q5, there were significant differences in the remaining variables. In the daily life pattern, NDG thought that their health was good, while LDG and MDG thought their health was good to some extent. On the other hand, HDG thought their health was poor. This trend was similarly observed in the questionnaire asking whether or not they had caught a cold. It was also found that HDG tended to sleep at a later time. In terms of meal frequency, NDG, LDG, and MDG showed little difference, whereas HDG had the highest meal frequency. In terms of exercise frequency per week, NDG was the highest, followed by LDG and MDG, sequentially. On the other hand, HDG showed the lowest exercise frequency.

### 3.3. Analysis of Body Composition

Though there was no significant difference in height, there was a significant difference in body weight among the four groups prior to the study. As shown in Table 2, HDG was significantly higher than the other three groups for body weight-related variables except for muscle mass. In the case of fat mass, NDG was the lowest, followed by MDG, LDG, and HDG. The post hoc test revealed that BMI, percent fat, and WHR of the HDG were significantly higher than those of other three groups. Meanwhile, muscle mass of the HDG was significantly lower than that of other three groups.

### 3.4. Analysis of Psychological Condition

The Cronbach’s α of the CES-D in this study was 0.795. When considering the total score of CES-D related to depression, there was a significant difference among groups. As shown in Figure 4, NDG (10.82 ± 4.62) was the lowest, followed by LDG (16.17 ± 5.89), MDG (19.93 ± 5.75), and HDG (32.50 ± 11.83). The questions that did not affect these results were CES-D 2, 6, 7, 8, and 16, whereas the rest showed significant differences among groups and had a significant effect on the total score. Characteristically, CES-D 8 and 16 were questions related to “I felt hope in the future” and “I just enjoyed life”, and all four groups showed similar results regardless of the degree of depression as shown in Appendix A of Appendix A. These results suggest that COVID-19 is undermining the hopes and joys of youth in the future. Similar to this context, CES-D 13, 14, and 15 were questions indicating “I spoke less than usual”, “I was lonely”, and “People did not feel friendly”, respectively. Although there were some differences between groups in NDG, LDG, and MDG in these three items, it can be observed that HDG shows a remarkably high score. In addition, it was confirmed that HDG gave more negative answers to other questions compared to the other three groups. Ultimately, although the CES-D sum scores of three groups were slightly different, this sum score of HDG was the highest (F = 39.964; *p* = 0.001; *η²* = 0.517) among the groups.

### 3.5. Analysis of Physical Fitness

As shown in Table 3, the physical fitness level of HDG was significantly lower than those of the other three groups although not significantly different in flexibility. In regards to strength and power, NDG was the highest, followed by LDG and MDG. The post hoc test revealed that HDG was significantly lower than the other groups. Characteristically, the case of cardioendurance was similar in NDG, LDG, and MDG but significantly lower only in HDG.

### 3.6. Analysis of Immunocytes

As shown in Figure 5 (left), NK cells showed significant differences among the four groups. Specifically, the NK cell level of NDG was 28.63 ± 6.60%, whereas those of LDG and MDG were 25.98 ± 6.91% and 22.14 ± 6.26%. The NK cell level of HDG was 18.92 ± 8.59%, and there was a significant difference (F = 7.637; *p* = 0.001; *η²* = 0.170) when comparing the four groups. It was found that the NK cell level of HDG was the lowest according to the post hoc test. As also shown in Figure 5 (right), the T-cell level of NDG was 61.54 ± 14.32%, whereas those of LDG and MDG were 60.47 ± 12.87% and 50.59 ± 12.25%, respectively. The T-cell level of HDG was 38.60 ± 18.12%, indicating the lowest level among the four groups. Eventually, there was a significant difference (F = 7.559; *p* = 0.001; *η²* = 0.168) when comparing the four groups with the post hoc test. These results indicate that the function of immune cells, which is an important defense system in the human body, decreases when feeling depressed or suffering mentally and shows that the more severe the depression, the worse the results can be.

## 4. Discussion

The main findings of this study are that the COVID-19 pandemic significantly affects the psychological state of male adolescents. A score exceeding 16 points in the CES-D questionnaire indicates depression. According to the results of this study, 39.0% of adolescents exceeded this score, and 7% of them felt depressed almost every day. According to the study results, the more severe the depression, the lower the scores in the questions discussing whether they consider themselves healthy and their degree of exposure to immune diseases such as colds. Moreover, adolescents involving in the HDG went to sleep late, ate more frequently, and exercised less. Specifically, the time for falling asleep as shown in Table 1 was 3.93 ± 0.80 and 3.83 ± 1.03 for NDG and LDG, almost between 11 and 12 o’clock, whereas for MDG and HDG, it was 4.60 ± 0.99 and 4.75 ± 0.46, showing almost no sleep after 12 o’clock in the evening. Diet frequency was similar in NDG, LDG, and MDG, but it was found that the eating frequency was significantly higher in HDG. Moreover, as for regular exercise, the higher the frequency of feeling depressed, the lower the rate of participation in exercise. That is, the NDG was 3.41 ± 1.61, which was the highest exercise frequency among the four groups, and the LDG and MDG were similar at 2.87 ± 1.60 and 2.67 ± 1.40, whereas the HDG was 1.38 ± 0.52, indicating that they did not exercise regularly for less than a day. It was confirmed that the body composition, physical fitness, and immune cell function of the adolescents investigated in this study were also negatively affected as they became psychologically depressed.

Recently, depression is emerging as a serious social problem among young people; 17% of depressed patients are in their 20s. According to a Korean Health Insurance Review and Assessment Service report [29], the number of depressed patients in their 20s was 595,724 in the first half of 2020, an increase of 5.8% from the previous year (563,239). The biggest social problem of depression is that the suicide rate of depressed patients is four times higher than that of the general population. Various psychological problems have been reported in countries heavily affected by the COVID-19 pandemic [10,30]. The social restrictions during the COVID-19 pandemic bring about lifestyle changes [31]. The most significant change was an increase in sedentary lifestyle, resulting in shorter physical activity time [32,33,34], which leads to lower physical fitness capacity and immunity [12]. This study found that sleep time and exercise frequency were shorter depending on the level of depression, as shown in Table 1. Physical fitness levels, cardiorespiratory endurance, strength, and power decreased as the degree of depression increased, and subjects with the highest level of depression had the lowest level of fitness, as shown in Table 3. Furthermore, HDG had the highest number of meals as well as body fat mass and WHR compared to the other three groups. In fact, obesity due to an increase in fat mass increases as physical activity decreases [35]. However, it is characterized by a continuous decrease in life satisfaction and happiness in the current COVID-19 era.

Compared to the various age groups, these psychological problems may appear more severe in adolescents, which is when the growth of the human body occurs most actively. Since it is an emotionally unstable period, depression, including antisocial risk, may appear [36]. Recently, more intense social distancing, closure of indoor cultural facilities, restrictions on outdoor public sports facilities, bans on religious gatherings, and the closure of movie theaters and karaoke rooms are bringing many changes to daily life in Korea. These tendencies have negative consequences not only on psychological factors but also on physical factors. These characteristics were found to be less well-managed in more depressed subjects than those with lower levels compared with pre-COVID-19. This was also confirmed in the interviews as follows:


*“Compared to pre-pandemic life, the situation has drastically changed because of online classes. I didn’t have to go to school, so I wasn’t nervous. During online classes, I couldn’t hear what they were talking about, so I just fell asleep.” Interview: a student (YOO)*



*“I slept a lot because I didn’t have to go to school. Of course, I slept even when it was time to eat. And, in the evening, I couldn’t sleep and had more time to play games. It seems that I ate ramen almost every day because I was hungry after playing games until dawn.” Interview: a student (KOO)*



*“At first, I didn’t go to school, so I felt good, and it was a different feeling. However, day by day, I became more anxious. I had to do something, but I did not know what to do”. Interview: a student (POO)*



*“When I first got into a lifestyle I wasn’t used to, I didn’t know what to do. As the number of corona patients continued to increase, I was afraid to go outside. When I saw the news, I was very afraid of the fact that the number of deaths was increasing while teenagers were not receiving vaccinations. That’s why I hated doing anything”. Interview: a student (LOO)*


As mentioned above, the daily lifestyle patterns in our society have negatively affected students who must spend most of their time at home due to the COVID-19 pandemic. If teenagers engage in physical activity, such as planks, walking, or running at home or outside, they can improve their body weight, physical fitness, and immunity. In this regard, Lee et al. [37] reported in one case study that performing elbow plank exercises for the elderly for 30 min a day, 5 days a week, for 4 weeks improved body composition, NK cells, and cytotoxicity. Similarly, Park et al. [38] also reported that a middle-aged adult who performed a vigorous elbow plank exercise for 20 min a day, 5 days a week, for 4 weeks showed increased physical fitness factors through improved skeletal muscle mass, basal metabolic rate, and body fat. They also argued that because performing plank exercises increased CD3, CD8, and CD56 but decreased CD4, CD4/CD8, and cytotoxicity, the exercise intensity should be adjusted to some extent. A recent study reported that exercising online or offline is helpful to maintain a level of physical fitness throughout the pandemic and is a great benefit to adolescents [39]. Seventeen-year-old students, named here as POO, COO, KOO, and COO, reflected on their engagement with physical activity compared to pre-COVID 19 times:


*“The online exercise was fun, though it was difficult at first. After exercising, I slept well and woke up in the morning. I thought it was better than doing nothing, so I continued”. Interview: a student (POO)*



*“My friends and I had a good time following the teacher’s exercises on our smartphones. It was fun... Anyways, it was good that we were the only ones doing things that other school kids couldn’t... It was a fun time.” Interview: a student (COO)*



*“I walked on the playground with my parents wearing a mask. I felt a lot different than when I walked without a mask... I had to take care of my health, so I tried to exercise regularly... Anyway, I tried to keep doing it, thinking that it is better than not doing it”. Interview: a student (KOO)*



*“In fact, we thought Corona was like a cold, so we used to run around the playground. Of course, I used to wear a mask over my chin... Most of the other children didn’t even cover their noses and mouths with a mask... It was also very stuffy, and the mask was wet with sweat. But playing soccer with friends is fun, so it’s good to pass the time.” Interview: a student (COO)*


The above interviews examined the changes in physical activity caused by the coronavirus, and they reflect the reality of the youth. The results from the three studies show that regular physical activity or exercise is beneficial for physical fitness, immunity, and mental health. Various circumstances are compounded by the prolonged fatigue of the COVID-19 pandemic. Currently, the number of patients infected by stealth Omicron is increasing rapidly in Korea. Ironically, while the number of infected and critically ill patients is increasing, the Ministry of Education is encouraging face-to-face classes, increasing student dissatisfaction and stress. Social isolation, health concerns, fear of infection, and weight gain due to lack of physical activity are also reported as causes of depression [40]. 

Korea’s fierce educational environment, which prioritizes academics over regular physical activity or exercise, further promotes inactive lifestyles. These results are likely to aggravate various chronic diseases among adolescents, including obesity, depression, and so on [11]. This study shows that students with higher levels of depression had the lowest strength and power, while other physical fitness factors were also significantly low. Ironically, it has been reported that, in general, overweight/obese adolescents have high strength, including grip strength. However, in the results of this study, the BMI of HDG, which showed high depression, corresponded to overweight or obesity, while grip strength, an index of muscle strength, was the lowest among the four groups. These results were inconsistent with the results of Mendoza-Muñoz et al. [41] but were consistent with those of Wind et al. [42]. Of course, it is a general theory that the thicker the cross-sectional area of the arm, the higher the muscle strength, but it is accepted that the muscle strength is low when the weight is controlled, and when inferring only from the results of this study, adolescents with depression are more likely to be obese or overweight, and it is thought that despite being able to exert sufficient muscle strength, they did not exert sufficient strength due to decreased willpower. This decreased willpower is also thought to be related to depression. Reduced muscle mass is a more serious consequence due to severe depression. When observing the skeletal muscle mass of the adolescents, NDG, LDG, and MDG were 25.89 ± 6.31 kg, 22.69 ± 4.82 kg, and 22.61 ± 5.60 kg, respectively, while HDG was 19.58 ± 5.60 kg. In this regard, Peake et al. [43] reported that a pro-inflammatory environment in the muscles may result in increased lymphocyte homing to the site of vaccine administration and enhanced antigen uptake and processing, making the initial phase of the immune response more efficient.

On the other hand, Hotamisligil [44] reported that the reduced muscle mass adversely affects cytokines and immune cells, resulting in immunotoxicity. Many studies have documented how abnormal responses from the immune system and the triggering of local inflammation are the first causes in the development of chronic diseases [12,45,46,47,48,49,50]. Like the results of the above studies, the muscle mass, exercise frequency, and high levels of fat observed in HDG were significantly lower than those of the other three groups, which suggests that mental depression could induce immune diseases. This study found that the percentage of NK cells responsible for innate immunity and T cells responsible for acquired immunity decreased in the subjects during the COVID-19 pandemic. It was also found that the decrease in immune cells could be affected by the severity of depression. These results were also confirmed in the interviews with two students in HDG as follows:


*“Because of the pandemic compared with the pre-COVID 19, I can’t go out often to meet my friends... I’m gaining more and more weight, my energy levels are low, and I don’t want to do anything.... Also, although it’s not Corona, I caught a cold easily and didn’t get better even if I took medicine...” Interview: a student (LOO)*



*“I couldn’t work out, so I tried to cut down on what I ate, but I gained weight. I need to study... I hate to do it... I fall asleep... but when I try to sleep, I can’t sleep.... Maybe it’s because I can’t move well... My fingers and knee joints are sore...” Interview: a student (POO)*


The current situation caused by the COVID-19 pandemic is adversely affecting the physical as well as the mental health of adolescents. In particular, the number of infected patients is increasing daily. If the situation does not improve, depression among adolescents will likely continue to rise. It is important for students to chat, play games, and engage in physical activity to prevent mental exhaustion, physical deterioration, and immune diseases. 

Ultimately, this study observed the psychological depression of adolescents living in the COVID-19 era. As a result, about 39.0% of the population was depressed, and 7% of adolescents experiencing severe depression were exposed to bad eating habits and lack of exercise. Consistent with these results, it was observed that as the depression worsened, physical strength decreased, and immune cell function decreased. Despite these findings, this study has several limitations as follows. First, there is a limit to generalizing the results of the study because only male students were sampled from a specific area and a specific school. Second, since the subjects of this study were 17 years old, it may be difficult to generalize and expand the results of this study to other age groups. Third, the daily life questionnaire and interview contents developed in this study had to be provided and collected through SNS, so there may be a lack of depth. Fourth, the body’s immune cells are innumerable. However, since only two types of immune cells were investigated in this study, it may be difficult to expand and interpret them. Fifth, this study has limitations in clarifying the causal relationship caused by the COVID-19 pandemic because there is no information on daily life, physical and psychological condition, and immune cells measured before the COVID-19 pandemic. Sixth, because of examining the CES among the 20 questions used in this study, it was generally found that a question #6 in the CES had the highest similarity to the 20 CES questions. For this reason, one question regarding their depression levels was used for grouping the subjects, and then, various variables were analyzed. However, since there is a problem in grouping the subjects with only one question, it is desirable to select a method that groups subjects using several questions in future research studies. At last, this study compares depression levels in students before and during the COVID-19 pandemic. In response to the pandemic, the Korean government implemented social distancing policies and restricted access to school facilities. This has prevented school-age students from having in-person meetings with teachers and friends as well as leading to changes in behavioral habits. It is reported that this phenomenon is exacerbating levels of depression in people of all ages. Once the COVID-19 pandemic ends, and social distancing policies are lifted, a follow-up study would be required to confirm whether depression levels will return to normal. Considering these limitations, further studies are encouraged to investigate the daily lifestyle pattern, psychophysical condition, and immunocytes with diverse demographic backgrounds and on multiple immunocyte tests.

## 5. Conclusions

This study found that the COVID-19 pandemic significantly affected the daily lifestyle pattern, psychophysical conditions, and immunocytes of adolescents. Regarding levels of depression, 39.0% of adolescents felt depressed, and 7% of them felt depressed almost every day. Overall, HDG considered themselves unhealthy and felt prone to immune diseases. HDG also went to sleep late and woke up at a consistent time, ate frequently, and exercised less. In the physical fitness factors, the cardiorespiratory endurance and strength of HDG were significantly lower than those of NDG, LDG, and MDG. Moreover, HDG had the worst body composition, including the lowest muscle mass. Finally, NK cells and T cells showed a significant difference among groups. The levels of HDG were significantly lower than those of the other groups. Therefore, it is suggested that appropriate measures are necessary to prevent adolescents from falling into depression.

## Figures and Tables

**Figure 1 healthcare-10-01152-f001:**
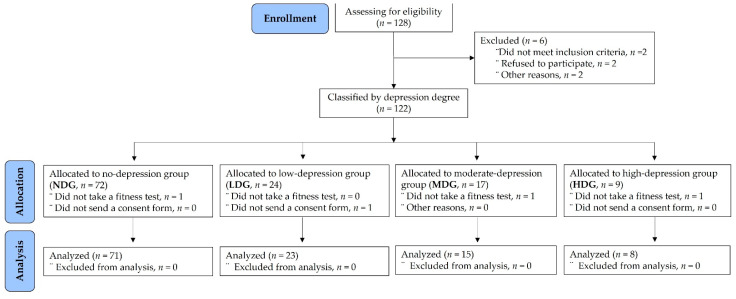
Participants’ allocation and analysis.

**Figure 2 healthcare-10-01152-f002:**
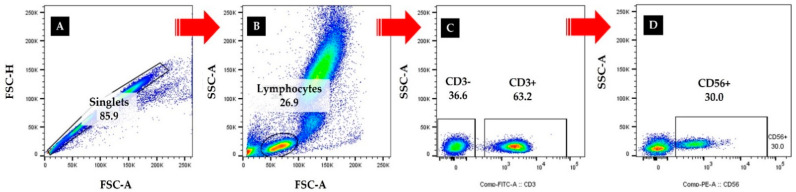
Flow cytometry for analyzing NK cells and T cells in the adolescents.Singlets were gated by area and height of forward scatter. A front-scattered light (FSC)-A and side-scattered light (SSC)-A plot was used to identify nucleated cells. (**A**) Singlets were gated, and doublets were excluded. (**B**) Lymphocyte gate was set based on the size and granularity of the cells. (**C**) CD3− and CD3+ were defined. (**D**) CD56+ was identified.

**Figure 3 healthcare-10-01152-f003:**
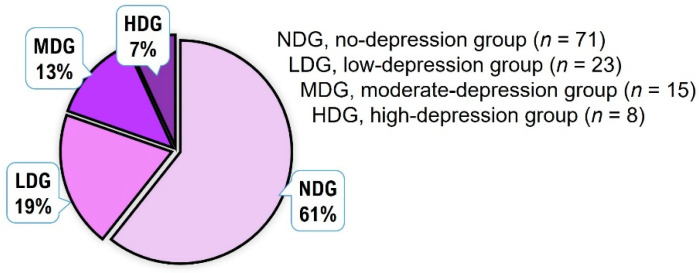
Frequency analysis of depression levels for all adolescents.

**Figure 4 healthcare-10-01152-f004:**
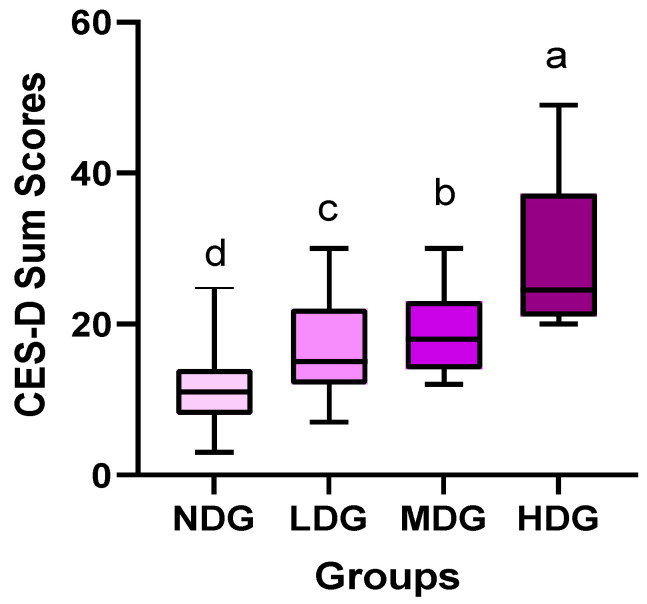
Comparative results of CES-D sum scores among four groups. Symbols ^a^, ^b^, ^c^, and ^d^ represent post hoc results from Bonferroni test.

**Figure 5 healthcare-10-01152-f005:**
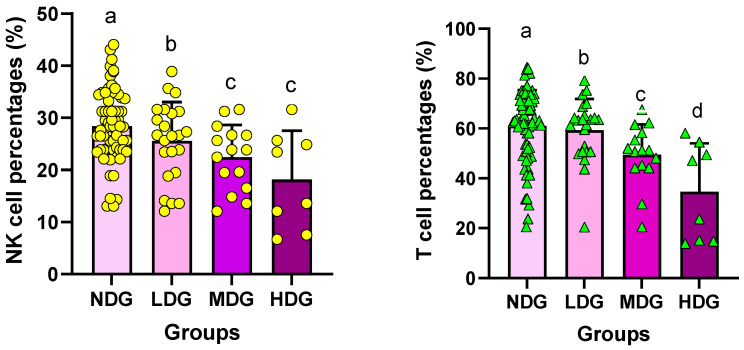
Differences of NK cells and T cells between the groups. Symbols ^a^, ^b^, ^c^, and ^d^ represent post hoc results from Bonferroni test.

**Table 1 healthcare-10-01152-t001:** Comparative results of daily life patterns among the four groups.

	Groups						
	NDG	LDG	MDG	HDG	F	*p*	*η²*
Q1. How healthy do you think you are?	3.19 ± 1.39 ^a^	2.76 ± 1.22 ^b^	2.36 ± 0.93 ^b^	1.57 ± 0.79 ^c^	4.603	0.005	0.114
Q2. Are you prone to respiratory diseases such as colds?	3.59 ± 1.35 ^a^	3.05 ± 1.20 ^a^	2.36 ± 0.74 ^b^	1.43 ± 0.53 ^c^	9.463	0.001	0.210
Q3. How many hours per day do you sleep on average?	3.49 ± 0.97	3.39 ± 1.12	3.00 ± 1.13	2.63 ± 1.51	2.126	0.101	0.054
Q4. What time do you usually sleep?	3.93 ± 0.80 ^b^	3.83 ± 1.03 ^b^	4.60 ± 0.99 ^a^	4.75 ± 0.46 ^a^	4.720	0.004	0.112
Q5. What time do you usually wake up?	4.90 ± 0.59	4.78 ± 0.80	4.60 ± 0.51	4.63 ± 0.52	1.541	0.208	0.040
Q6. How many meals do you eat per day?	2.86 ± 0.62 ^b^	2.65 ± 0.57 ^b^	2.93 ± 0.80 ^b^	4.25 ± 0.71 ^a^	11.487	0.001	0.235
Q7. How many days a week do you exercise for at least 30 min a day?	3.41 ± 1.61 ^a^	2.87 ± 1.60 ^b^	2.67 ± 1.40 ^b^	1.38 ± 0.52 ^c^	6.393	0.001	0.146

All data represent mean ± standard deviation. Symbols ^a^, ^b^, and ^c^ represent post hoc results from Bonferroni test. NDG, no-depression group; LDG, low-depression group; MDG, moderate-depression group; HDG, high-depression group.

**Table 2 healthcare-10-01152-t002:** Comparative results of body composition among the four groups.

	Groups						
	NDG	LDG	MDG	HDG	F	*p*	*η²*
Height (cm)	164.99 ± 13.97	163.67 ± 7.17	163.14 1 ± 8.17	167.15 ± 7.54	0.268	0.849	0.007
Body weight (kg)	60.04 ± 13.28 ^b^	57.85 ± 11.19 ^b,c^	54.12 ± 10.67 ^c^	71.76 ± 15.88 ^a^	5.505	0.018	0.085
Muscle mass (kg)	25.89 ± 6.31 ^a^	22.69 ± 4.82 ^b^	22.61 ± 5.60 ^b^	19.58 ± 5.06 ^c^	18.857	0.001	0.336
Fat mass (kg)	14.59 ± 5.06 ^d^	16.90 ± 5.63 ^b,c^	15.18 ± 6.50 ^b,c^	24.18 ± 11.13 ^a^	5.870	0.001	0.136
BMI (kg/m²)	21.29 ± 3.53 ^b^	21.34 ± 3.40 ^b^	21.17 ± 3.11 ^b^	26.10 ± 4.26 ^a^	4.691	0.004	0.112
Percent fat (%)	21.45 ± 7.09 ^c^	23.04 ± 4.76 ^b^	23.40 ± 6.95 ^b^	29.13 ± 4.15 ^a^	2.920	0.037	0.073
WHR	0.83 ± 0.05 ^c^	0.85 ± 0.06 ^b^	0.84 ± 0.06 ^b,c^	0.91 ± 0.07 ^a^	3.670	0.014	0.090

All data represent mean ± standard deviation. Symbols ^a^, ^b^, ^c^, and ^d^ represent post hoc results from Bonferroni test. BMI, body mass index; WHR, waist/hip ratio; NDG, no-depression group; LDG, low-depression group; MDG, moderate-depression group; HDG, high-depression group.

**Table 3 healthcare-10-01152-t003:** Comparative results of physical fitness among the four groups.

	Groups						
	NDG	LDG	MDG	HDG	F	*p*	*η²*
Cardiorendurance (reps.)	47.35 ± 19.95 ^a^	43.05 ± 17.73 ^b^	45.57 ± 19.96 ^a,b^	22.00 ± 10.91 ^c^	4.694	0.004	0.112
Flexibility (cm)	11.49 ± 9.50	12.55 ± 9.34	9.79 ± 15.11	5.30 ± 13.59	0.979	0.405	0.026
Strength (kg)	29.46 ± 9.73 ^a^	25.14 ± 6.94 ^b^	22.86 ± 5.76 ^b,c^	19.29 ± 4.99 ^c^	10.696	0.001	0.223
Power (m)	193.16 ± 27.64 ^a^	190.76 ± 35.62 ^a^	181.14 ± 31.75 ^b^	158.86 ± 22.84 ^c^	3.905	0.011	0.095

All data represent mean ± standard deviation. Symbols ^a^, ^b^, and ^c^ represent post hoc results from Bonferroni test. NDG, no-depression group; LDG, low-depression group; MDG, moderate-depression group; HDG, high-depression group.

## Data Availability

All the data are available in this paper.

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
