# Peer review of "Daily Life Patterns, Psychophysical Conditions, and Immunity of Adolescents in the COVID-19 Era: A Mixed Research with Qualitative Interviews by a Quasi-Experimental Retrospective Study"

_healthcare, 2022, doi:10.3390/healthcare10061152_

Round 1
Reviewer 1 Report
A very interesting study. Scientifically sound and well presented.
Could you please add a mention of a need for a follow up study to see if depression symptoms persevered or whether they diminished as learning was switched back on on site settings?
Could you also please add a mention of how the study was conducted under unusual circumstances that in themselves might have contributed to the depression and such a mental state might not be a permanent state of mind for the participants?
The questions used in the interview were leading and not neutral. Could you please add an explanation as to why this was so?
Author Response
Answers to 1st reviewer’s comments
Thank you for your kind advice and comments for publication in Healthcare. We revised our manuscript as per your comments. We represented the specific modifications in response to the comments by blue letters in our manuscript. We sincerely appreciate your comments because your constructive feedback greatly improves our manuscript.
A very interesting study. Scientifically sound and well presented.
Q1: Could you please add a mention of a need for a follow up study to see if depression symptoms persevered or whether they diminished as learning was switched back on site settings?
#Response 1: Thank you for your valuable comment. Based on your suggestion, the necessity of a follow-up study has been added to the Discussion section.
Line 549:
“… At last, this study compares depression levels in students before and during the COVID-19 pandemic. In response to the pandemic, the Korean government implemented social distancing policies and restricted access to school facilities. This has prevented school-age students from having in-person meetings with teachers and friends, as well as leading to changes in behavioral habits. It is reported that this phenomenon is exacerbating levels of depression in people of all ages. Once the COVID-19 pandemic ends and social distancing policies are lifted, a follow-up study would be required to confirm whether depression levels will return to normal.”
Q2: Could you also please add a mention of how the study was conducted under unusual circumstances that in themselves might have contributed to the depression and such a mental state might not be a permanent state of mind for the participants?
#Response 2: Thank you for pointing this out. Based on your comments, the following information has been inserted into the research method.
Line 86:
“The Korean government looked at the trend of newly infected COVID-19 cases and adjusted restrictions accordingly. In line with this, school authorities decided whether in-person classes would take place and required students to wear masks at school. For this study, a research plan was made according to what we believed was likely to be the circumstances at the start of the study. Upon approval from the IRB committee, the experiment was carried out.”
Q3: The questions used in the interview were leading and not neutral. Could you please add an explanation as to why this was so?
#Response 3: Thank you for your comment and suggestion regarding the interview questions. When we first started this study, the research model focused solely on 'quantitative research', but we wanted to convey the direct feelings of the students who were going through the COVID-19 pandemic while checking the overall research status after completing the preliminary research. Based on your suggestion, these contents were adapted and inserted into the research method as follows.
Line 180:
“When we first planned this study, the research model focused entirely on 'quantitative research', but we wanted to convey the direct feelings of the students going through the COVID-19 pandemic after completing the preliminary survey.”
Thank you for your comments.
Sincerely,
May 30, 2022
Reviewer 2 Report
Daily Life Patterns, Psychophysical Conditions, and Immunity of Adolescents in the COVID-19 Era: A Mixed Research with Qualitative Interviews
This study incorrectly describes its research design as a 1) randomized experiment and 2) a prospective cross-sectional study. This are major and significant limitations of the study.
The design of this study is not a randomized experiment because the 117 subjects were assigned to one of four conditions (depression groups – NDG, LDG, MDG, HDG) based on their score on one single item of the CES-D scale. To be randomized means study participants are randomly put into one of the four conditions, but people can’t be randomly assigned to have no depression (NDG group) or low depression (LDG group), etc. Hence, the authors have very skewed sample sizes within each group (71, 23, 15, and 8). The largest group is the NDG (n=71) because most subjects had no depression. This is not a randomized experiment. It is also problematic to categorize “depression” based on one single item from a larger scale. I am not a clinical psychologist, but those I have worked with used scales (with multiple items) to diagnose depression (or any mental condition).
The authors are also incorrect to describe their study as a “prospective” study. It is a cross-sectional study, with some retrospective data. Authors asked study participants to recall their lives pre-Covid.
The description for the sample for the study is confusing. In the abstract, the “subjects included 17-year-old male adolescents.” Inn section 2.1, the “average age of the subjects was 17 years old.” Precision in describing the data is important.
The participants go on to describe power-analysis for their study (which is not randomized) – but experiments, particularly those with physiological data, are being pre-registered in advance of data collection. Researchers calculate the sample size a priori of the data collection. This is to prevent p-hacking, which has become a huge issue in psychology. Based on the author’s write-up on the power analysis, it seems to over-sampled and recruited more study participants than they needed. This is problematic in terms of p-hacking.
This study has some interesting physiological data, and I wondered if there was some way to re-frame the study and design, but it is extremely problematic to categorize depression based on one single-item from a larger questionnaire. Authors should use more standardized scales to detect depression.
-END-
Author Response
Answers to 2nd reviewer’s comments
Thank you for your kind advice and comments for publication in Healthcare. We revised our manuscript as per your comments. We represented the specific modifications in response to the comments by blue letters in our manuscript. We sincerely appreciate your comments because your comments make our manuscript better.
Q1: This study incorrectly describes its research design as a 1) randomized experiment and 2) a prospective cross-sectional study. This are major and significant limitations of the study.
#Response 1: Thank you for bringing this error to our attention. We addressed this mistake and replaced the incorrectly stated description of our study, "a quasi-experimental retrospective study." The title, as well as the research method, were all revised accordingly.
Title was changed to, “Daily life patterns, psychophysical conditions, and immunity of adolescents in the COVID-19 era: A mixed research with qualitative interviews by a quasi-experimental retrospective study”
“2.2. Experimental design
This was a quasi-experimental retrospective study that investigated both quantitative research analysis and qualitative interview content to understand how the COVID-19 pandemic affects adolescents' daily lives, psychophysiological conditions, and immunocyte function.”
Q2: The design of this study is not a randomized experiment because the 117 subjects were assigned to one of four conditions (depression groups – NDG, LDG, MDG, HDG) based on their score on one single item of the CES-D scale. To be randomized means study participants are randomly put into one of the four conditions, but people can’t be randomly assigned to have no depression (NDG group) or low depression (LDG group), etc. Hence, the authors have very skewed sample sizes within each group (71, 23, 15, and 8). The largest group is the NDG (n=71) because most subjects had no depression. This is not a randomized experiment. It is also problematic to categorize “depression” based on one single item from a larger scale. I am not a clinical psychologist, but those I have worked with used scales (with multiple items) to diagnose depression (or any mental condition).
#Response 2: Thank you for pointing this out. As mentioned above, our experimental design was modified to "a quasi-experimental retrospective study", and the word 'random' was deleted from all contents. You pointed out that it is problematic to categorize “depression” based on one single item from a larger scale. However, as a result of examining the 20-item CES, this study showed a high degree of similarity to item #6 in the CES. For this reason, we hope that you understand that this is a grouping based on a question that the students themselves answered as being depressed. This will also be included in the limitations toward the end of the discussion.
Line 543:
“Sixth, because of examining the CES among the 20 questions used in this study, it was generally found that a question #6 in the CES had the highest similarity to the 20 CES questions. For this reason, one question regarding their depression levels was used for grouping the subjects, and then various variables were analyzed. However, since there is a problem in grouping the subjects with only one question, it is desirable to select a method that groups subjects using several questions in future research studies.”
Q3: The authors are also incorrect to describe their study as a “prospective” study. It is a cross-sectional study, with some retrospective data. Authors asked study participants to recall their lives pre-Covid.
#Response 3: Thank you for your comment. Based on what you pointed out, we modified the relevant parts of our manuscript by describing our study as "a quasi-experimental retrospective study".
Q4: The description for the sample for the study is confusing. In the abstract, the “subjects included 17-year-old male adolescents.” Inn section 2.1, the “average age of the subjects was 17 years old.” Precision in describing the data is important.
#Response 4: Thank you for pointing this out to us. Since all the subjects are male students of the same grade, '17-year-old male adolescents' is correct. Therefore, the sentence that you pointed out was modified as follows.
Line 110:
“All subjects were 17 years old.”
Q5: The participants go on to describe power-analysis for their study (which is not randomized) – but experiments, particularly those with physiological data, are being pre-registered in advance of data collection. Researchers calculate the sample size a priori of the data collection. This is to prevent p-hacking, which has become a huge issue in psychology. Based on the author’s write-up on the power analysis, it seems to over-sampled and recruited more study participants than they needed. This is problematic in terms of p-hacking.
#Response 5: Thank you for your insightful feedback. It may come as no surprise that the sampling in this study appears to be inflated. In fact, in our country, all school-aged students are required to have their health, as well as physical fitness, examined at every grade level. Therefore, while the number of samples obtained by considering the experimental design of this study as G*power was 84, the number of students who were obligated to take the test in order to participate in this study had to exceed 84. To avoid any misunderstanding regarding the above, the following sentences have been added into the data processing method as follows.
Line 115:
"The number of samples obtained by considering the experimental design of this study as G*power was 84, but the number of samples was rather high as there were 117 students who had to undergo mandatory health and physical examinations."
Q6: This study has some interesting physiological data, and I wondered if there was some way to re-frame the study and design, but it is extremely problematic to categorize depression based on one single-item from a larger questionnaire. Authors should use more standardized scales to detect depression.
#Response 6: Thank you for your comment. CES-D is a questionnaire that is used to identify depression and consists of 20 items. Among them, it would not be an exaggeration to say that question #6 is most representative of the entire CES-D because it is a question that directly asks about depression levels. In addition to the one question, this study also considered the total CES-D score, so we considered that there was no problem in grouping students in this study. However, in respect of the reviewers' comments, we would like to include the above as a limitation of the study at the end of the discussion.
Line 543 to 549:
“Sixth, because of examining the CES among the 20 questions used in this study, it was generally found that a question #6 in the CES had the highest similarity to the 20 CES questions. For this reason, one question regarding their depression levels was used for grouping the subjects, and then various variables were analyzed. However, since there is a problem in grouping the subjects with only one question, it is desirable to select a method that groups subjects using several questions in future research studies.”
Thank you for your comments.
Sincerely,
May 30, 2022
Reviewer 3 Report
The topic of this research is interesting and of importance. But there are some problems associated with the research method. The details are as follows:
- The subjects are not “randomly” assigned into the four groups. They are assigned into the groups based on the CES-D score. Throughout the manuscript, the terms “random”, “randomly”, or “randomized” are misused.
- The recruitment process of the subjects seems to be flawed. It should be that consent form is signed before any other procedures were performed. But in this study, there was a subject who did not sign the consent form. In addition, most of the subjects were minor. It should be that the guardians or parents who sign the consent form, not the students themselves.
- The study design states that “ … to understand how the COVID-19 pandemic affects adolescents’ daily lives, …..”. In the qualitative part, four questions were asked regarding COVID-19. The effects of COVID-19 can be addressed. But the quantity questionnaires were collected during the pandemic as a snapshot of the ongoing condition. There is no comparison before or after the pandemics. Thus, the quantitative part of the study design cannot assess the effect of COVID-19. It only describes a state of outcome “during” the pandemic. As a result, any sentences in this manuscript related to the effects of COVID on quantitative outcomes such as mental, health, lifestyle, immune factors should be rephrased.
- The presentation of the result of Table 1 should be improved. Table 2 provides comparisons between the four groups for each question. However, it makes more sense to translate the numbers into plain language. For example, sleep hours could be directly described as percentage of respondents sleep less than 5 hours, or between 5-6 hours, etc. This is especially important when explaining the results in the discussion section. The first paragraph of Discussion section should be significantly rewritten.
- Table 4 describes the results for each CES-D question. Are the details of each questions relevant? Is each question of special importance? If the content of the detailed questions were not discussed in the manuscript, the result from each question is not relevant to be presented in detail either. The sum of the 20 questions, on the other hand, could be sufficient. Alternatively, the 20 questions could be categorized into some subsections and discussed by the nature of subgroups. In short, Table 4 provide meaningless information to the readers.
- Cronbach alpha is for testing internal consistency. Its usage in Table 1 is very confusing. Why each question would generate a Cronbach alpha value? If it is for testing the consistency across subject, the application is very wrong.
- In sum, it is awkward to combine both the quantitative and qualitative parts into this single study since these two parts are aiming at different angles. The former is analyzing the snapshot outcome within the pandemic and the latter is about the comparisons before and during the pandemic. The former contains comparisons of subjects across groups with different level of depression. But the latter does not link explicitly with the depression level.
Author Response
Answers to 3rd reviewer’s comments
Thank you for your kind advice and comments for publication in Healthcare. We revised our manuscript as per your comments. We represented the specific modifications in response to the comments by blue letters in our manuscript. We sincerely appreciate your comments because your comments make our manuscript better.
The topic of this research is interesting and of importance. But there are some problems associated with the research method. The details are as follows:
- The subjects are not “randomly” assigned into the four groups. They are assigned into the groups based on the CES-D score. Throughout the manuscript, the terms “random”, “randomly”, or “randomized” are misused.
#Response 1: Thank you for bringing this to our attention. We addressed this mistake and replaced the incorrectly stated description of our study, "a quasi-experimental retrospective study." The title, as well as the research method, were all revised accordingly.
Title was changed to, “Daily life patterns, psychophysical conditions, and immunity of adolescents in the COVID-19 era: A mixed research with qualitative interviews by a quasi-experimental retrospective study”
“2.2. Experimental design
This was a quasi-experimental retrospective study that investigated both quantitative research analysis and qualitative interview content to understand how the COVID-19 pandemic affects adolescents' daily lives, psychophysiological conditions, and immunocyte function…”
- The recruitment process of the subjects seems to be flawed. It should be that consent form is signed before any other procedures were performed. But in this study, there was a subject who did not sign the consent form. In addition, most of the subjects were minor. It should be that the guardians or parents who sign the consent form, not the students themselves.
#Response 2: Thank you for your insightful comment. In our country of South Korea, students in all school are required to have their health, as well as physical fitness, examined at every grade level. Since the above process has been considered compulsory, consent forms were obtained only from students. According to your comments, we accepted your proposal and obtained informed consent from the guardians who volunteered to be our subjects, and we have inserted the following related sentences into the text (Line 79).
Line 93 to 96:
"Although the tests performed in this study were compulsory tests performed by schools, we obtained informed consents from their parents as students were considered a vulnerable group."
- The study design states that “ … to understand how the COVID-19 pandemic affects adolescents’ daily lives, …..”. In the qualitative part, four questions were asked regarding COVID-19. The effects of COVID-19 can be addressed. But the quantity questionnaires were collected during the pandemic as a snapshot of the ongoing condition. There is no comparison before or after the pandemics. Thus, the quantitative part of the study design cannot assess the effect of COVID-19. It only describes a state of outcome “during” the pandemic. As a result, any sentences in this manuscript related to the effects of COVID on quantitative outcomes such as mental, health, lifestyle, immune factors should be rephrased.
#Response 3: Thank you for pointing this out. Based on your comments, 'after the pandemics' has been changed to 'during the pandemic'.
- The presentation of the result of Table 1 should be improved. Table 2 provides comparisons between the four groups for each question. However, it makes more sense to translate the numbers into plain language. For example, sleep hours could be directly described as percentage of respondents sleep less than 5 hours, or between 5-6 hours, etc. This is especially important when explaining the results in the discussion section. The first paragraph of Discussion section should be significantly rewritten.
#Response 4: Thank you for your observation and suggestion. In response, Table 1 and Table 2 were improved. Following the recommendations of other reviewers, Table 1 was converted to Table S1 and sent to supplementary materials. Therefore, Table 2 in the existing paper became Table 1. Please refer to the improved tables in the manuscript. And, as shown in Table 1, the part you recommended to express as 'sleep hours' does not show a significant difference between the 4 groups, so it is difficult to interpret. However, there was a statistically significant difference in Q4's sleep time (What time do you usually sleep?), which was interpreted at the top of the discussion as follows.
Line 374:
“Moreover, adolescents involving in the HDG went to sleep late, ate more frequently, and exercised less. Specifically, the o'clock for falling asleep as shown in Table 1 was 3.93 ± 0.80 and 3.83 ± 1.03 for NDG and LDG, almost between 11 and 12 o'clock, whereas, for MDG and HDG, it was 4.60 ± 0.99 and 4.75 ± 0.46, showing almost no sleep after 12 o'clock in the evening.”
- Table 4 describes the results for each CES-D question. Are the details of each questions relevant? Is each question of special importance? If the content of the detailed questions were not discussed in the manuscript, the result from each question is not relevant to be presented in detail either. The sum of the 20 questions, on the other hand, could be sufficient. Alternatively, the 20 questions could be categorized into some subsections and discussed by the nature of subgroups. In short, Table 4 provide meaningless information to the readers.
#Response 5: Thank you for your comments. Table 4 was deleted according to your suggestion, and only the total scores for the 20 questions in the questionnaire were reinterpreted. And according to the recommendations of other reviewers, Table 4 was converted to Table S2 and sent to supplementary materials.
- Cronbach alpha is for testing internal consistency. Its usage in Table 1 is very confusing. Why each question would generate a Cronbach alpha value? If it is for testing the consistency across subject, the application is very wrong.
#Response 6: Thank you for informing us of this misapplication. In response, the Cronbach alpha value for each individual item was deleted, and only the Cronbach alpha value for the total items was presented.
- In sum, it is awkward to combine both the quantitative and qualitative parts into this single study since these two parts are aiming at different angles. The former is analyzing the snapshot outcome within the pandemic and the latter is about the comparisons before and during the pandemic. The former contains comparisons of subjects across groups with different level of depression. But the latter does not link explicitly with the depression level.
#Response 7: Thank you for your constructive comments. In fact, when we first planned this study, the research model focused only on 'quantitative research', but we wanted to convey the direct feelings of the students who were experiencing the COVID-19 pandemic while checking the overall research status after completing the preliminary research. Based on your opinion, these contents were adapted as follows and inserted into the research method.
Line 180 to 183:
“When we first planned this study, the research model focused only on 'quantitative research', but we wanted to convey the direct feelings of the students who were experiencing the COVID-19 pandemic after completing the preliminary research.”
Thank you for your comments.
Sincerely,
May 30, 2024
Reviewer 4 Report
I find the study very interesting and the topic is quite topical, as it tries to explore the consequences that covid is leaving or is going to leave in our society. However, the study is limited to one part of the population, boys, so I think it would be much more interesting to assess the inclusion of the female sex to know how it affects them as well. The study would be much more powerful as comparisons could be made. So my main suggestion to the authors is to justify the choice of this population and the discrimination against the same population of the opposite sex.
In addition, I would like to make some minor comments:
Put in the inclusion criteria everything they considered, which then causes them to eliminate participants, for example: signing the informed consent.
Why were only boys included? Justification should be given as to why only this population was included, why girls were not included, was it an exclusion criterion to be a girl, why?
Why were participants who had suffered from coronovirus excluded, it may be that some participants had suffered from coronovirus and were asymptomatic and are included, do they refer to participants who have had symptoms or who have had the disease, if they have had the disease, what is the reason for the exclusion, how does it affect their results, and if so, what is the reason for the exclusion?
It is recommended to include the sections "design", "ethical approval" and "sample calculation", in the same order, before "participants".
Table 1 is not relevant, it could be included as supplementary material.
In relation to this sentence "In the general social science field, the acceptance criterion for accreditation of reliability is considered to be 0.6 or higher", include citation or include citation in statistical analysis.
Include height in table 3.
The HDG group seems to have a high degree of overweight and/or obesity, the strength fitness results are lower than the rest of the groups, how do you justify this, as the scientific literature amply exposes that the results in handgrip tend to be in favour of overweight and/or obese participants:
- Deforche B, Lefevre J, De Bourdeaudhuij I, Hills AP, Duquet W, Bouckaert J. Physical fitness and physical activity in obese and nonobese Flemish youth. Obesity research. 2003;11(3):434-41.
- Lad UP, Satyanarayana P, Shisode-Lad S, Siri CC, Kumari NR. A study on the correlation between the body mass index (BMI), the body fat percentage, the handgrip strength and the handgrip endurance in underweight, normal weight and overweight adolescents. Journal of clinical and diagnostic research: JCDR. 2013;7(1):51.
- Mendoza-Muñoz, M., Adsuar, J. C., Pérez-Gómez, J., Muñoz-Bermejo, L., Garcia-Gordillo, M. Á., & Carlos-Vivas, J. (2020). Influence of body composition on physical fitness in adolescents. Medicina, 56(7), 328.
Author Response
Answers to 4th reviewer’s comments
Thank you for your kind advice and comments for publication in Healthcare. We revised our manuscript as per your comments. We represented the specific modifications in response to the comments by blue letters in our manuscript. We sincerely appreciate your comments because your comments make our manuscript better.
Q1: I find the study very interesting and the topic is quite topical, as it tries to explore the consequences that covid is leaving or is going to leave in our society. However, the study is limited to one part of the population, boys, so I think it would be much more interesting to assess the inclusion of the female sex to know how it affects them as well. The study would be much more powerful as comparisons could be made. So my main suggestion to the authors is to justify the choice of this population and the discrimination against the same population of the opposite sex.
#Response 1: Thank you for your kind comments. As we were looking for a school that regularly conducts a health checkup and a physical examination for the students every year, we came to investigate a school made up of only male students. As the same researcher, we ask for your generosity. However, we suggested the restrictions related to gender are explained at the end of the discussion as follows.
On line 532 to 534:
“First, there is a limit to generalizing the results of the study because only male students were sampled from a specific area and a specific school.”
In addition, I would like to make some minor comments:
Q2: Put in the inclusion criteria everything they considered, which then causes them to eliminate participants, for example: signing the informed consent.
#Response 2: Thank you for what the reviewer has pointed out the comments. We have corrected the points you brought to our attention. In other words, all the factors that the subjects considered were inserted into the inclusion criteria as follows.
Line 119 to 122:
“Students who were required to undergo a mandatory health and physical examination every year were included in this study. The inclusion criteria also required that the participants had not received treatment or medication known to affect mental status and body composition.”
Q3: Why were only boys included? Justification should be given as to why only this population was included, why girls were not included, was it an exclusion criterion to be a girl, why?
#Response 3: Thank you for your questions and request for clarification. As we were looking for a school that regularly conducts student checkups and physical examinations every year, we came to investigate a school made up of only male students. Restrictions related to gender are explained at the end of the discussion as follows.
Line 534 to 536:
“First, there is a limit to generalizing the results of the study because only male students were sampled from a specific area and a specific school.”
Q4: Why were participants who had suffered from coronovirus excluded, it may be that some participants had suffered from coronovirus and were asymptomatic and are included, do they refer to participants who have had symptoms or who have had the disease, if they have had the disease, what is the reason for the exclusion, how does it affect their results, and if so, what is the reason for the exclusion?
#Response 4: Thank you for your keen observation and suggestions for further explanation. We have addressed what you pointed out as follows. It is thought that infection with the corona virus causes significant changes in immune cell function, which can act as an extra variable, not only in physical strength, but also in psychological aspects, so it was excluded from this study. Totally, we inserted above contents in the text as follows.
Line 123 to 127:
“Students who had or had been infected with the coronavirus or corona variant virus were also excluded from the study. It is thought that infection with the corona virus causes significant changes in immune cell function, which can act as an extra variable, not only in physical fitness, but also in psychological aspects, so it was excluded from this study.”
Q5: It is recommended to include the sections "design", "ethical approval" and "sample calculation", in the same order, before "participants".
#Response 5: Thank you for pointing this out. In response, the methods of this study were rearranged in the order of "design," "ethical approval," "sample calculation," and "participants" as follows.
“2. Materials and Methods
2.1. Experimental design
This was a quasi-experimental retrospective study that investigated both quantitative research analysis and qualitative interview content to understand how the COVID-19 pandemic affects adolescents' daily lives, psychophysiological conditions, and immunocyte function. All data in this study were acquired from September 6, 2021 to October 6, 2021. The participants we recruited for this study were male middle school students. Since the participants’ conditions before the COVID-19 pandemic could not be quantitatively grasped, they were asked during the qualitative interview to make a comparison of their daily life and psychophysiological conditions before and during the COVID-19 pandemic.
2.2. Ethical approval
The Korean government looked at the trend of newly infected COVID-19 cases and adjusted restrictions accordingly. In line with this, school authorities decided whether in-person classes would take place and required students to wear masks at school. For this study, a research plan was made according to what we believed was likely to be the circumstances at the start of the study. Upon approval from the IRB committee, the experiment was carried out. Prior to the study, the participants and their parents received detailed explanations regarding the study procedures and were then asked to complete a questionnaire, have a blood sample taken, and complete a physical fitness test. Although the tests performed in this study were compulsory tests performed by schools, we obtained informed consents from their parents as students were considered a vulnerable group. All the subjects, including the researchers, did not know which group the participants belonged to. This study started after receiving IRB approval (2-1040781-A-N-012020 085HR) and the data of all participants were kept confidential and used for research purposes only. After thoroughly checking the social distancing status when students were able to go to school, only students who had normal body temperature and no COVID-19 symptoms gathered in the experimental research center. Daily life patterns that can affect the mind and body included self-health and disease status, sleep and wake-up time, meal frequency, and exercise. For the questionnaire investigating psychological state, CES-D was used to gauge the depression levels of the participants, and physical condition was measured as a health-related physical fitness component. The natural killer (NK) and T cells were identified through blood tests.
2.3. Subjects
This study recruited 128 male students in their third year of an all-boys middle school in Korea who expressed their intention to participate in this study. All subjects were 17 years old. The sample size using G*Power (v. 3.1.9.7, Heinrich-Heine-University Software, Germany) was calculated by adding the number of subjects required in the analysis of covariance (ANCOVA), considering a priori effect size of f2 (V) = 0.40 (large effect), α error probability = 0.05, power (1-β error probability) = 0.95, number of groups = 4, and numerator difference = 1. The number of samples obtained by considering the experimental design of this study as G*power was 84, but the number of samples was rather high as there were 117 students who had to undergo mandatory health and physical examinations.“
Q6: Table 1 is not relevant, it could be included as supplementary material.
#Response 6: Thank you for your suggestion. According to the reviewer's opinion, Table 1 was included as supplementary material. And the existing Table 1 was modified to Table S1.
Q7: In relation to this sentence "In the general social science field, the acceptance criterion for accreditation of reliability is considered to be 0.6 or higher", include citation or include citation in statistical analysis.
#Response 7: Thank you for pointing this out. References to "In the general social science field, the acceptance criterion for accreditation of reliability is considered to be 0.6 or higher" were found and accompanied with citation as below.
Line 284:
In the general social science field, the acceptance criterion for accreditation of reliability is considered to be 0.6 or higher [28].
[28] Hair, J.F.; Black, W.C.; Babin, B.J.; Anderson, R.E. Multivariate Data Analysis (7th ed.). Pearson Education Limited. 2014. Retrieved from https://www. pearson.com/us/higher-education/program/ Hair-Multivariate-Data-Analysis-7thEdition/PGM263675.html.
Q8: Include height in table 3.
#Response 8: Thank you for your suggestion. In response, height was added to Table 3.
Q9: The HDG group seems to have a high degree of overweight and/or obesity, the strength fitness results are lower than the rest of the groups, how do you justify this, as the scientific literature amply exposes that the results in handgrip tend to be in favour of overweight and/or obese participants:
- Deforche B, Lefevre J, De Bourdeaudhuij I, Hills AP, Duquet W, Bouckaert J. Physical fitness and physical activity in obese and nonobese Flemish youth. Obesity research. 2003;11(3):434-41.
- Lad UP, Satyanarayana P, Shisode-Lad S, Siri CC, Kumari NR. A study on the correlation between the body mass index (BMI), the body fat percentage, the handgrip strength and the handgrip endurance in underweight, normal weight and overweight adolescents. Journal of clinical and diagnostic research: JCDR. 2013;7(1):51.
- Mendoza-Muñoz, M., Adsuar, J. C., Pérez-Gómez, J., Muñoz-Bermejo, L., Garcia-Gordillo, M. Á., & Carlos-Vivas, J. (2020). Influence of body composition on physical fitness in adolescents. Medicina, 56(7), 328.
#Response 9: Thank you for pointing this out in your comments. Based on your opinion, two references and the results of this study on the relationship between obesity and muscle strength have been inserted into the discussion section as follows.
On line 483 to 494:
“Ironically, it has been reported that, in general, overweight/obese adolescents have high strength including grip strength. However, in the results of this study, the BMI of HDG, which showed high depression, corresponded to overweight or obesity, while grip strength, an index of muscle strength, was the lowest among the four groups. These results were inconsistent with the results of Mendoza-Muñoz et al. [41] but were consistent with those of Wind et al. [42]. Of course, it is a general theory that the thicker the cross-sectional area of the arm, the higher the muscle strength, but it is accepted that the muscle strength is low when the weight is controlled. And when inferring only from the results of this study, adolescents with depression are more likely to be obese or overweight, and it is thought that despite being able to exert sufficient muscle strength, they did not exert sufficient strength due to decreased willpower. This decreased willpower is also thought to be related to depression.”
[41] Mendoza-Muñoz, M.; Adsuar, J.C.; Pérez-Gómez, J.; Muñoz-Bermejo, L.; Garcia-Gordillo, M.Á.; Carlos-Vivas, J. Influence of body composition on physical fitness in adolescents. Medicina (Kaunas) 2020, 56, 328. doi: 10.3390/medicina56070328.
[42] Wind, A.E.; Takken, T.; Helders, P.J.; Engelbert, R.H. Is grip strength a predictor for total muscle strength in healthy children, adolescents, and young adults? Eur. J. Pediatr. 2010, 169, 281-287. doi: 10.1007/s00431-009-1010-4.
Thank you for your comments.
Sincerely,
May 30, 2024
Round 2
Reviewer 2 Report
Even with the corrections made in the second round, the use of a single-item to detect depression - an method that is very controversial in psychological studies both in clinical settings and in research settings - is extremely problematic. The most common survey method to detect depression is Patient Health Questionnaire-9 which contains 9 questions. Increasingly though researchers are using biomarkers to detect depression. Nevertheless, even without biomarkers, authors should use more than one single item to detect depression.
Reviewer 4 Report
The authors responded to all comments.
Nothing to add.